# Gestational Pesticide Exposure and Child Respiratory Health

**DOI:** 10.3390/ijerph17197165

**Published:** 2020-09-30

**Authors:** Robyn Gilden, Erika Friedmann, Katie Holmes, Kimberly Yolton, Yingying Xu, Bruce Lanphear, Aimin Chen, Joseph Braun, Adam Spanier

**Affiliations:** 1Department of Family and Community Health, University of Maryland School of Nursing, Baltimore, MD 21201, USA; 2Office of Research and Scholarship, University of Maryland School of Nursing, Baltimore, MD 21201, USA; friedmann@umaryland.edu (E.F.); kholmes@umaryland.edu (K.H.); 3Department of Pediatrics, Cincinnati Children’s Hospital Medical Center, University of Cincinnati College of Medicine, Cincinnati, OH 45229, USA; Kimberly.Yolton@cchmc.org (K.Y.); Yingying.Xu@cchmc.org (Y.X.); 4Department of Health Sciences, Simon Fraser University, Burnaby, BC V5A 1S6, Canada; bruce_lanphear@sfu.ca; 5Department of Biostatistics, Epidemiology and Informatics, University of Pennsylvania Perelman School of Medicine, Philadelphia, PA 19104, USA; aimin.chen@pennmedicine.upenn.edu; 6Department of Epidemiology, Brown University, Providence, RI 02912, USA; joseph_braun_1@brown.edu; 7Department of Pediatrics, Division of General Pediatrics, University of Maryland School of Medicine, Baltimore, MD 21201, USA; aspanier@som.umaryland.edu

**Keywords:** organophosphates, pyrethroids, gestational exposure, children, wheeze

## Abstract

Background: Childhood wheeze may be related to pesticide exposure, and diet and genetics (Paroxonase; *PON1*) may modify the effects of exposure. Methods: We analyzed data from the HOME Study, a prospective pregnancy and birth cohort, to examine the association of gestational urinary organophosphate (OP) and pyrethroid (3PBA) metabolite concentrations with child wheeze, forced expiratory volume in one second (FEV1) at ages 4 and 5 years, and wheeze trajectory patterns through age 8 years. Results: Among 367 singletons, the frequency of wheeze ranged from 10.6% to 24.1% at each measurement age. OP and 3PBA metabolite concentrations were not associated with wheeze at 8 years or from birth to 8 years, but there were three significant interactions: (1) maternal daily fruit and vegetable consumption (less than daily consumption and increasing 3PBA was associated with wheeze at age 8 years, OR = 1.40), (2) maternal *PON1*_-*108*_ allele (CT/TT genotypes and high DE was associated with wheeze at age 8 years, OR = 2.13, 2.74) and (3) *PON1_192_* alleles (QR/RR genotypes with higher diethylphosphate (DE) and dialkyl phosphate (DAP) were associated with wheeze at age 8 years, OR = 3.84). Pesticide metabolites were not consistently related to FEV1 or wheeze trajectory. Conclusions: Gestational OP and 3PBA metabolites were associated with child respiratory outcomes in participants with maternal dietary and genetic susceptibility.

## 1. Introduction

The prevalence of asthma in children has increased over the past four decades. In 2017, the lifetime prevalence among all children younger than 18 was 8.4%; however, the prevalence was higher among African American (12.6%), Hispanic (11.3%), and low-income children (11.0%) [1]. Uncontrolled asthma can diminish children’s quality of life and increase school absenteeism [2], leading to about 15 million missed school days per year [3]. Asthma results in increased urgent care visits [4], emergency department visits, hospitalizations [5], and health care costs [4,5,6]. Asthma is the third leading cause of hospitalization for children under 15 [3], and the estimated annual cost of medical expenses for a child with asthma is USD 1000 compared to a little over USD 600 for one without asthma [7].

The increase in the prevalence of asthma over recent decades cannot be attributed solely to genetic changes; therefore, it is likely that environmental exposures are contributing to this increase [8]. Several studies have implicated exposure to pesticides in the development of wheeze, a precursor to asthma [9,10,11]. Studies of occupationally exposed adults [12] and pregnant women suggest a potential link between exposure to some pesticides and respiratory disorders [10,13,14,15]. One study in particular identified a potential link between gestational exposure to a pyrethroid metabolite, 3-Phenoxybenzoic acid (3PBA), and child cough at age 5 to 6 years [16].

Prior investigations have not considered genetic susceptibility, which may be critical because genes can modify the effects of some exposures. Paraoxonase (*PON1*) is known to detoxify organophosphate (OP) pesticides [17], and depending on the gene polymorphism, can modify the effects of OP exposure [18]. Two specific *PON1* polymorphisms (*_−_*_108 and 192_) have been implicated as potential modifiers of the association between OP pesticide exposure and poor birth and childhood health outcomes, including reduced birth weight, shortened length of gestation, and neurobehavioral issues [19,20,21,22]. Other studies have investigated the link between diet and asthma exacerbation. One study [23] assessed the modification of fruit and vegetable intake on inflammation in asthmatic children in Mexico City and found that a higher intake of fresh produce and closely following the Mediterranean diet had a protective effect on the lung health of children with asthma. A 2017 systematic review [24] of 58 studies also supported the same conclusion of protective effect fruit consumption against asthma. The purpose of our study was to examine the association between prenatal pesticide exposure and respiratory health in children, including genetic polymorphisms involved in pesticide metabolism and maternal vegetable and fruit consumption as potential modifying factors.

## 2. Methods

We conducted a secondary data analysis within the Health Outcomes and Measures of the Environment (HOME) Study, a prospective pregnancy and birth cohort that followed mothers and their children in the greater Cincinnati, Ohio, USA metropolitan area from the second trimester of pregnancy [25]. The HOME Study was designed to assess the relationship between low-level environmental chemical exposures and many aspects of child health, including child respiratory outcomes. Inclusion criteria during enrollment of pregnant women included: (1) 16 ± 3 weeks gestation, (2) ≥18 years old, (3) living in a home built before 1978 (this was to focus the cohort on potential lead exposure), (4) no history of HIV infection, and (5) not taking medications for seizure or thyroid disorders. A total of 468 pregnant women were enrolled between March 2003 and January 2006, and 390 mothers delivered live-born singletons who were followed from birth to age 8 years. The study was reviewed and approved by the Institutional Review Board at Cincinnati Children’s Hospital Medical Center and the University of Maryland [26]. All mothers provided informed consent for themselves and their children.

### 2.1. Pesticide Exposure Assessment

A detailed description of the pesticide exposure assessment has previously been reported [27]. In short, mothers provided urine samples twice during pregnancy, at 16 and 26 weeks gestation (16 ± 3 weeks; 26 ± 4 weeks). We stored samples in polypropylene containers at −20 °C before being shipped to and analyzed by the Centers for Disease Control and Prevention (CDC) for metabolites of OPs (diethyldithiophosphate (DEDTP), diethylphosphate (DEP), diethylthiophosphate (DETP), dimethyldithiophosphate (DMDTP), dimethylphosphate (DMP), dimethylthiophosphate (DMTP)) and 3PBAs using lyophilization with gas chromatography-tandem mass spectrometry and isotope dilution quantification [28]. We measured urinary creatinine concentrations to account for urine dilution.

The primary independent variables were gestational urinary metabolite concentrations of OP and 3PBA pesticides. For the OPs, we created three aggregated measures for each participant. We calculated diethyl phosphate (DE) as the sum of DEDTP, DEP, and DETP in molar concentration at each time point (16 and 26 weeks). We calculated dimethyl phosphate (DM) as the sum of DMDTP, DMP, and DMTP at each time point (16 and 26 weeks). We calculated combined dialkyl phosphate (DAP) as the sum of DE and DM. Then, we standardized the DE, DM, and DAP variables for creatinine. The creatinine standardized sums were then log_2_-transformed to approximate normality. The two transformed values (16-week and 26-week) were averaged to create final OP pesticide metabolite concentrations for each of the metabolite groups. We calculated the final 3PBA concentration using a similar approach.

### 2.2. Paraoxonase (PON1) Genotype Assessment

Paraoxonase (PON1) is an enzyme that detoxifies several classes of pesticides, such as OPs and 3PBA. The genotypes analyzed in this study were *PON1_−108_*_(CC, CT, TT)_ and *PON1_192_*
_(QQ, QR, RR)._ The process for genotyping is described in detail elsewhere [21]; however, frozen archived cord blood was used to extract and analyze DNA following standard protocols. Briefly, these included using the Applied Biosystem predesigned TaqMan assays (Applied Biosystems, Carlsbad, CA, USA) for rs705379 (108C/T) and rs662 (192Q/R) with 15 ng genomic DNA. The protocol was performed following the manufacturer’s instructions using the 384-well format, including 16 blanks and 4 sets of 2 controls for quality control purposes [21].

### 2.3. Respiratory Outcomes

The primary respiratory outcome variable was derived using a parent survey about wheezing episodes (yes/no) based on the National Health and Nutrition Examination Survey (NHANES). These were collected every 6 months, from children ages 6 months to 5 years, then at ages 6 years and 8 years (Has (child’s name) had wheezing or whistling in his/her chest in the last 6 months?” [29]). Participant data were marked as missing if the participant did not attend the research visit, did not complete the survey question, or was lost in follow-up.

We also measured Forced Expiratory Volume in 1 s (FEV1) using a portable device (PiKo-1; nSpire Health, Inc., Longmont, CO, USA) during study visits at ages 4 and 5 years following American Thoracic Society standardized procedures [30]. Briefly, at the study visit in the research clinic, children were coached by research assistants to exhale at maximum capacity and duration into the PiKo-1 device. Their nose was occluded with a clip to prevent air leakage. Three attempts were made per child, and the maximum acceptable FEV1 recorded was used as the FEV1 for each subject. We then calculated the percentage predicted FEV1 (PPFEV1), adjusted for child height, and multiplied by 0.9 for those of black race using standard approaches at the time the HOME study was conducted [31].

### 2.4. Covariates

We considered including covariates in our analyses based on prior studies and biologic plausibility. We first conducted bivariate analyses between the covariates and the exposure variables as well as the outcome variables. Any variable that was significantly associated in bivariate analyses (*p* < 0.2) with the predictor or outcome was considered for inclusion in multivariable models. We evaluated maternal factors including age at delivery, maternal race (white versus non-white), marital status (not married versus married), educational level (some high school, some college or 2 year degree, bachelor’s degree), employment status (no versus yes), parity (0, 1, >1), alcohol intake (no versus yes), and daily fruit and vegetable consumption during pregnancy (less than daily versus daily). We examined child variables, including birth weight and sex. We also considered variables which could be related to respiratory health and wheeze, such as serum cotinine measured in maternal samples collected at 16 weeks and 26 weeks gestation using high performance liquid chromatography-tandem mass spectrometry (HPLC-MS/MS) [32,33]. The cotinine assay limit of detection (LOD) was 0.015 ng/mL, with a coefficient of variation (CV) ranging from 3 to 4% at high concentrations (1 ng/mL) to 10% at low concentrations (0.1 ng/mL) [34].

### 2.5. Data Analysis

We used means and standard deviations to describe the central tendency and dispersion of pesticide metabolites and PPFEV1 scores. We calculated the proportion of children reporting wheeze at each assessment time point. We analyzed the association between log_2_ transformed pesticide metabolite concentrations and the following respiratory outcomes: (1) wheeze at age 8 years (within the previous 6 months), (2) wheeze in the previous 6 months across all study visits from ages 1 to 8 years (up to 12 time points), (3) PPFEV1 at ages 4 and 5 years, and (4) patterns of wheeze over time.

We used multiple logistic regression to evaluate the association of gestational pesticide metabolite concentrations with child wheeze at age 8 years, and we examined interactions of exposures with maternal *PON1_−108_* and *PON1_192_* alleles and daily fruit and vegetable consumption during pregnancy. We conducted separate analyses for DE, DM, DAP, and 3PBA concentrations. Covariates included were child’s birth weight and sex, mother’s age at delivery, employment, education, alcohol consumption, parity, and daily fruit and vegetable consumption. Cotinine did not meet criteria for inclusion as a covariate. In additional analysis, we stratified by maternal race and examined the association of *PON1* alleles and daily fruit and vegetable consumption within each group.

We used generalized linear mixed models with logit link to evaluate the prospective association of gestational pesticide metabolite concentrations with each child’s wheeze over the 8 years (at each time point; time treated continuously). Covariates remained the same as in the multiple logistic regression analyses. We used linear regression to evaluate the association of gestational pesticide metabolite concentrations with each child’s PPFEV1 at ages 4 and 5 years in separate analyses, including the same set of covariates.

We used linear regression to evaluate the association of gestational pesticide metabolites with each child’s PPFEV1 at ages 4 and 5 years in separate analyses. We examined interactions by maternal *PON1_−108_*, *PON1_192_* alleles.

Finally, we identified children’s wheeze patterns with a discrete mixture model based trajectory analysis for clustering of longitudinal data series [35]. Four patterns of children’s wheeze trajectories were identified: persistent wheezers (those who wheezed across most time points), early wheezers (those who wheezed in early time points but did not wheeze in later time points), late wheezers (those who did not wheeze in early time points but did wheeze in later time points), and unlikely wheezers (those who did not wheeze across most time points). Multinomial logistic regression was used to determine if gestational pesticide metabolite levels predicted wheeze trajectory group membership. We conducted separate analyses for each OP exposure variable (DE, DM, and DAP) as well as the 3PBA exposure variable with persistent wheezers as the reference. We also examined the interaction of pesticide metabolites with *PON1* genotypes and daily fruit and vegetable consumption during pregnancy.

Stata version 12 was used for regression (reg), logistic regression (logit), and generalized linear models (genlin) as well as for the trajectory (traj) analysis. Estimates for longitudinal graphs were produced in SPSS version 25 to enable inclusion of random effects (genlinmixed). We used a two-sided probability of 5% determined statistical significance for all analyses.

## 3. Results

Of the 390 singletons who were followed up between birth and 8 years, 367 had wheeze outcome data and were included in the current study (Table 1). Over half of the children were white (61.7%) and female (54.4%). Most (80.1%) of the mothers reported eating fruit and vegetables at least once per day during pregnancy. For the *PON1_−108_* gene polymorphism, most of the mothers (46.1%, *n* = 140/304) had the CC genotype. For the *PON1_192_* gene polymorphism, the largest percentage was genotype QR (41.4%, *n* = 132/319). The 23 subjects without any wheeze data across all time points were more likely to have indicators of low socioeconomic status (race, maternal education, marital status, income, birth weight, and gestational age), but they did not differ in the measures of pesticide exposure (results not shown).

Geometric means of maternal pesticide metabolite concentrations at 16 and 26 weeks gestation varied by race (Table 2). 

Based on t-tests, compared to white mothers, non-white mothers had significantly higher 3PBA concentrations at both 16 and 26 weeks (*p* = 0.005 and *p* = 0.002, respectively), DM concentrations at 26 weeks (*p* = 0.003), and DAP concentrations at 26 weeks, (*p* = 0.008).

At 48 months (*n* = 155), the average PPFEV1 score was 64.3% (SD = 21.7%). At 60 months (*n* = 197), the average PPFEV1 score was 68.8% (SD = 20.6%).

### 3.1. Wheeze at Age 8 Years

In multivariable analyses, none of the maternal OP (DE, DM, DAP) or 3PBA metabolite concentrations during pregnancy were associated with the presence of wheeze in the 6 months prior to the 8 year visit (Appendix A). Daily maternal fruit and vegetable consumption interacted with 3PBA metabolite concentrations. Children of mothers who did not consume fruit and vegetables daily during their pregnancy had greater odds of wheezing at age 8 years as their mother’s gestational 3PBA metabolites increased (interaction OR = 1.35; 95% CI 1.013, 1.815 with a two-fold increase in 3PBA; *p* = 0.038). For children of mothers who consumed fruits and vegetables at least daily during pregnancy, higher 3PBA metabolite concentration was associated with lower odds of wheezing (Figure 1).

There were significant interactions between maternal *PON1*_−*108*_ alleles and gestational OP metabolite concentrations with odds of wheeze at age 8 years. For children of mothers with *PON1_−108_* CC alleles (*n* = 153), as concentrations of DM metabolites increased, odds of child wheeze increased. For children of mothers with *PON1_−108_* CT alleles (*n* = 131), as concentrations of DM metabolites increased, odds of child wheeze decreased (interaction OR = 1.721, 95% CI 1.010, 2.932, *p* = 0.046; Figure 2a). The relationship of mother’s DM metabolite concentration to odds of child wheeze at age 8 years did not differ significantly for children of mothers with *PON1_−108_* CC and TT alleles (Figure 2a). The relationship of DAP metabolite concentrations and *PON1_−108_* CC/CT alleles to odds of child wheeze at age 8 years tended to follow the same pattern as for DM metabolite concentration (interaction OR = 1.88, 95% CI 0.980, 3.740, *p* = 0.057; Figure 2b). For children of mothers with *PON1_−108_* TT alleles (*n* = 39), as DAP metabolite concentrations increased, odds of child wheeze at age 8 years tended to increase at a higher rate than for children of mothers with CC alleles (interaction OR = 2.809, 95% CI 0.849, 9.295, *p* = 0.091; Figure 2b). The relationship of DE metabolite concentration and *PON1_−108_* CC alleles to odds of child wheeze at age 8 years did not differ significantly from those with CT alleles (Figure 2c). For children of mothers with *PON1_−108_* TT alleles, as DE metabolite concentrations increased, odds of child wheeze at age 8 years tended to increase at a higher rate than for children of mothers with CC alleles (interaction OR = 2.94, 95% CI 0.095, 1.214, *p* = 0.096; Figure 2c). The relationship of 3PBA metabolite concentrations to odds of child wheeze at age 8 years did not differ significantly between *PON1_−108_* CC and *PON1_−108_* CT alleles (Figure 2d); it tended to differ between *PON1_−108_* CC and *PON1_−108_* TT alleles (interaction OR = 1.56, 95% CI 0.95, 0.990, 2.58, *p* = 0.078, Figure 2d). As maternal 3PBA metabolite concentrations increased, odds of child wheeze at age 8 years increased for children of mothers with *PON1_−108_* CC alleles and decreased for children of mothers with *PON1_−108_* TT alleles.

There were significant interactions between maternal *PON1_192_* alleles and gestational OP metabolite concentrations as predictors of odds of wheeze at age 8 years. There was an interaction between *PON1_192_* and gestational DE and DAP metabolite concentrations with wheeze at age 8 years. There was a greater relationship of both metabolites to child wheeze at age 8 years among those with QR/RR alleles (OR = 2.315, 95% CI 1.283, 4.172, *p* = 0.005, Figure 3) compared with those who had QQ alleles. The relationship of DAP metabolite concentration to wheeze was 1.5 times greater for those with *PON1_192_* QR/RR alleles (OR = 2.5, 95%, CI 1.062, 5.917, *p* = 0.036) than for those with the QQ allele (Figure 3). Additionally, the interaction of *PON1_192_* and the DM metabolite concentration neared significance (OR = 2.04, 95% CI 0.780, 4.405, *p* = 0.079). There was no significant interaction of *PON1_192_* alleles and 3PBA metabolites in relation to child wheeze at age 8 years.

Race-Stratified Analysis: We stratified by race due to differences in exposure (Table 2) and found results that were similar to the main analysis. Among whites, there was no interaction between maternal gestational 3PBA metabolites and *PON1_−108_* alleles with child wheeze at age 8 years, but the associations between OP metabolites and child wheeze trended toward significance. The relationship of higher maternal DE and DAP metabolite concentration with higher odds of wheezing for those with CC alleles versus those with TT alleles among white children was similar to what we found in the unstratified analysis (DE: interaction OR 4.860, 95% CI 0.805, 29.348, *p* = 0.085; DAP: interaction OR 3.859, 95% CI 0.910, 16.364, *p* = 0.067). Additionally, for white mothers, there was a significant interaction between *PON1_192_* alleles and the mother’s gestational DE metabolite concentrations with the child’s wheeze at age 8 years (interaction OR = 3.817, 95% CI = 1.574, 9.259, *p* = 0.003). Increased gestational concentrations of DE were associated with a greater increase in odds of wheeze among children of mothers with QR/RR alleles than among those with QQ alleles (interaction OR = 1.56, 95% CI 0.95, 0.990, 2.58, *p* = 0.078; Figure 4). The numbers of non-whites in the sample and the distribution of alleles did not lend themselves to most separate analyses, leading to non-convergence or extreme ORs due to over-specification.

### 3.2. Wheeze across Childhood (Repeated Measures over 8 Years)

Gestational OP metabolite concentrations were not associated with the presence of wheeze within the past six months during the first 8 years of the child’s life (Appendix A). There was no interaction of race, daily fruit and vegetable consumption, or *PON1* genotype with gestational OP metabolite concentrations in relation to child wheeze. There was an interaction, however, of maternal daily fruit and vegetable consumption during pregnancy with gestational 3PBA metabolite concentrations related to the child’s wheeze. For mothers who consumed less than one fruit and vegetable serving a day during pregnancy, higher gestational 3PBA metabolites were associated with higher odds of child wheeze, with the opposite effect for children of mothers who consumed one or more fruit and vegetable servings per day (interaction OR = 1.35; 95% CI 1.085, 1.678; *p* = 0.007, Figure 5).

### 3.3. FEV1

In bivariate correlation analysis, maternal OP metabolite concentrations during pregnancy were associated with a child’s respiratory function at age 4 but not 5 years. Correlations of DM concentration with PPFEV1 at age 4 years were small but significant (PPFEV1: r = 0.192, *p* = 0.017), and correlations of DAP metabolite concentrations were similar (PPFEV1: r = 0.207, *p* = 0.01). Concentration of 3PBA was not correlated with PPFEV1 at age 4 years (PPFEV1: r= −0.056, *p* = 0.489) (Appendix A)

In multiple regression analysis, OP metabolite concentrations in mothers during pregnancy were associated with a child’s respiratory function at age 4 but not 5 years; concentration of 3PBA was not associated with a child’s respiratory function. DM and DAP metabolite concentrations predicted PPFEV1 at age 4 years (DM: β = 2.878, 95% CI 0.311, 5.445; *p* = 0.028; DAP: β = 3.553, 95% CI 0.579, 6.527, *p* = 0.02).

There was a significant interaction of 3PBA metabolites with maternal *PON1_−108_* alleles for PPFEV1 at 4 years; interactions of 3PBA metabolites with *PON1_192_* were not significant. The differences in curves between the CC allele group (*n* = 66) and CT allele group (*n* = 52) were significant (PPFEV1: interaction β = −4.334, 95% CI −8.497, −0.168 *p* = 0.042), while those between the CC and TT allele groups (*n* = 18) were not (PPFEV1: interaction β = −9.521, 95% CI −22.797, 3.756, *p* = 0.157, Figure 6).

### 3.4. Trajectory Analysis

Almost half (*n* = 169, 46%) of the children were unlikely to have ever wheezed. This category also includes children with missing data precluding the use of the never wheezer label. The other three trajectory groups included 34.8% (*n* = 128) early wheezers, 8.2% (*n* = 30) late wheezers, and 10.9% (*n* = 40) persistent wheezers. In multivariable logistic regressions where data were missing on some of the covariates, the frequencies of wheeze trajectories were: persistent wheezers (*n* = 22, 11.4%), early wheezers (*n* = 44, 36.5%), late wheezers (*n* = 15, 15.1%), and unlikely wheezers (*n* = 94, 37.0%). When logistic regression analyses were used to examine the association of maternal gestational pesticide metabolites with the wheeze trajectory group, gestational pesticide metabolite concentrations were not significantly associated with persistent, early, or late wheeze trajectory groups when compared with the unlikely wheeze trajectory group.

## 4. Discussion

In this analysis, gestational exposure to OP and 3PBA pesticide metabolites were not independently associated with child wheeze at age 8 years or repeated wheeze from ages 6 months to 8 years. Two of the OP pesticide metabolites (DM and DAP) were associated with higher PPFEV1 at age 4 but not 5 years, while 3PBA was not associated with either. No significant relationships were found between gestational pesticide metabolite levels and child wheeze trajectory groups. There were several significant interactions in these analyses. Maternal daily consumption of fruit and vegetables during pregnancy interacted with gestational 3PBA metabolite concentrations where less than daily consumption and increasing concentrations of 3PBA increased the odds of child wheeze. Genetic variations in *PON1* impacted the odds of child wheeze with gestational exposure to OPs, mostly for the DE metabolite, but not for 3PBA pesticides.

Neither pesticide class was consistently related to any respiratory outcomes independently, but specific exposures among certain subgroups may be associated with child respiratory health. Gestational OP and 3PBA metabolites were not associated with wheeze at age 8 years, but we found a significant interaction of maternal daily fruit and vegetable intake and 3PBA metabolites. Among mothers with low fruit and vegetable intake, increasing 3PBA concentrations were associated with increased odds of wheeze at age 8 years. This supports preliminary findings from a recent systematic review and meta-analysis that found potential protection of fruit and vegetable consumption against wheeze and asthma [24]. Socio-economic status may also impact respiratory health [36].

While exposure to gestational OP and 3PBA pesticide metabolites was not associated with wheeze at age 8 years, there was a significant interaction based on *PON1* gene variants in subpopulations. Children born to mothers with the CC genotype of *PON1_−108_* and who had increasing gestational DM, DAP, and DE metabolite concentrations had increased odds of wheeze at age 8 years. Mothers with the QR/RR genotype of *PON1_192_* and who had increasing gestational DE and DAP metabolite concentrations had children with increased odds of wheeze at age 8 years. For both pesticide groups, those mothers with the CC polymorphism and higher metabolite levels had children with greater odds of wheeze at age 8 years. *PON1_192_* also interacted with the relationship of the gestational DE metabolite to child wheezing at age 8 years; the group of mothers with the QR/RR polymorphism and higher DE metabolites had children with greater odds of wheeze at age 8 years. A previous study [21] also examined the *PON1_192_* polymorphisms and determined that the heterozygous allele and increasing levels of gestational OP metabolites were related to a decrease in length of gestation and birth weight, but the investigators did not explore gestational 3PBA metabolite exposures. Additional research is needed to explore the role of these genes in mediating respiratory health risks of gestational exposure to OP and 3PBA pesticides.

Similar to the association of gestational OP and 3PBA pesticide metabolites with wheeze at age 8 years, there was no main effect of pesticide metabolite concentration with repeated child wheeze. In additional analyses, there was an interaction of the association of 3PBA concentration with repeated wheeze. Among mothers with low maternal daily fruit and vegetable intake, higher 3PBA metabolite concentrations were associated with increased odds of child wheeze over the 8 years. These results support the sparse but growing evidence of the relationship of 3PBA exposure to poor childhood respiratory health [16]. In the cited study, investigators measured permethrins and piperonyl butoxide (PBO, a 3PBA synergist and potential permethrin marker) in the third trimester and then again in indoor air when the children were at age 5–6 years. Gestational PBO was associated with increased odds of cough (OR (95% CI): 1.27 (1.09–1.48), *p* < 0.01; *n* = 217), but age 5–6 year PBO or permethrins were not associated with cough.

While there was no independent relation of gestational OP metabolites with wheeze at age 8 years or wheeze at any time over 8 years, paradoxically, gestational OP metabolites were positively correlated with a child’s PPFEV1 at age 4 years but not at age 5 years. In other studies, chlorpyriphos exposure was associated with decreased lung function in adolescent Egyptian agriculture workers [10,37]. Our finding of a seemingly protective effect might reflect the subpopulation of children able to perform the spirometry maneuver at that age. Additional research is needed to verify this finding. There was also a significant interaction of 3PBA metabolite concentrations and *PON1_−108_* in relation to PPFEV1 at age 4. Mothers with the CT allele who had higher 3PBA metabolite concentrations had children with lower PPFEV1 concentrations at age 4 year than mothers with the CC allele. The proposal that genetic variation may be involved in airway damage was first suggested in a study by Seo et al. [38]. They also investigated *PON1_192_* and found that in smokers with certain SNPS in PON1, there was a lower FEV1 suggesting a possible role of PON1 in mediating environmental exposure and pulmonary outcomes.

The trajectory analysis demonstrated four categories of wheeze that were similar to those previously reported: persistent wheezers, early wheezers, late wheezers, and unlikely wheezers [39]. We found no difference in maternal gestational OP or 3PBA metabolite concentrations in the persistent, early, or late wheeze trajectory groups compared with the unlikely wheeze group. This suggests that gestational levels may not be associated with specific wheeze patterns, and is consistent with a previous report that OP concentrations were not related to wheeze based on a cross-sectional analysis of NHANES data [9]. Our failure to find an association could also be related to the smaller sample size for the trajectory analysis.

This study has several limitations. First, maternal daily fruit and vegetable consumption and the outcome variable of child wheezing were based on maternal reports and may have been affected by recall bias. However, the respiratory outcome data were collected prospectively, which enhances our confidence in the survey questions. Second, although using biomarkers of exposure augment the survey data, there are limitations of using DAPs as a measure of OP metabolites. Given their short half-life and possible resulting exposure misclassification [40] as well as their non-specific nature, DAPs metabolites might not indicate exposure to the parent compound, but rather direct exposure to DAPs. Third, the participants were limited to English-speaking families, which may affect the generalizability of our findings. Finally, the results reflect simultaneous exposures to other toxicants and factors that were not accounted for in this analysis but could have unmeasured effects on respiratory health.

## 5. Conclusions

In this prospective pregnancy and birth cohort study, we did not find an overall association between gestational OP or 3PBA metabolites and child respiratory health outcomes. However, we found a link between gestational exposure to OP and 3PBA pesticides metabolites and respiratory outcomes in certain subpopulations. Some *PON1* variants (CT, TT alleles of *PON1_−108_*, and QR/RR allele of *PON1_192_*) enhanced negative respiratory outcomes related to pesticide exposures. There was also an interaction between daily fruit and vegetable consumption and gestational exposure to pesticides in relation to the odds of wheeze; increased daily fruit and vegetable consumption appeared to mollify the increased odds of wheeze associated with increased 3PBA exposure. In addition, OP metabolites were associated with an increase in PPFEV1 at 4 years of age. Lastly, trajectory analysis found no association between gestational pesticide exposure and early, late, or persistent wheeze groups when compared to unlikely wheezers. These preliminary findings suggest potential targets for intervention and the need for additional research among subgroups to help improve child respiratory health. For example, we could intervene with gestational education to improve daily fruit and vegetable intake and reduce pesticide exposure—particularly among those with low intake or with higher risk PON1 alleles.

## Figures and Tables

**Figure 1 ijerph-17-07165-f001:**
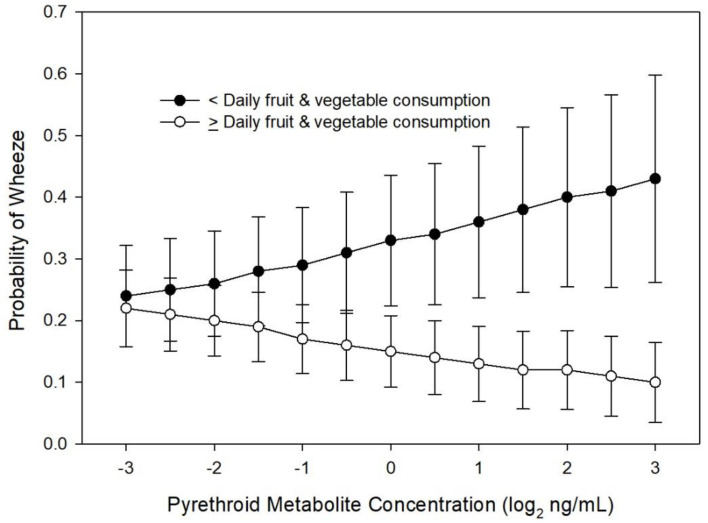
Probability of child wheeze at age 8 years related to maternal pyrethroid (3-Phenoxybenzoic acid, 3PBA) metabolite concentration and maternal daily fruit and vegetable consumption during pregnancy (*n* = 225).

**Figure 2 ijerph-17-07165-f002:**
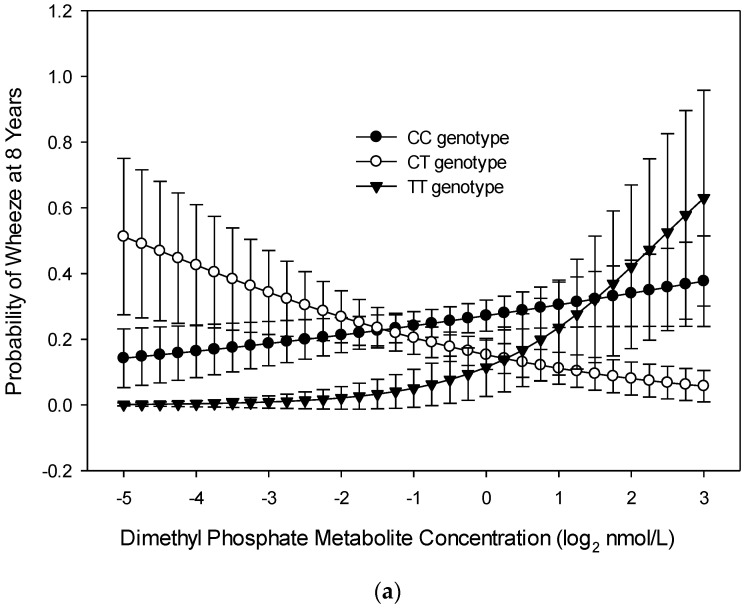
Probability of child wheeze at age 8 years related to maternal *PON1_−108_* genotype and (**a**) dimethyl phosphate (DM) metabolite concentration, (**b**) dialkyl phosphate (DAP) metabolite concentration, (**c**) diethyl phosphate (DE) concentration, and (**d**) pyrethroid (3-Phenoxybenzoic acid, 3PBA) metabolite concentration during gestation (*n* = 224).

**Figure 3 ijerph-17-07165-f003:**
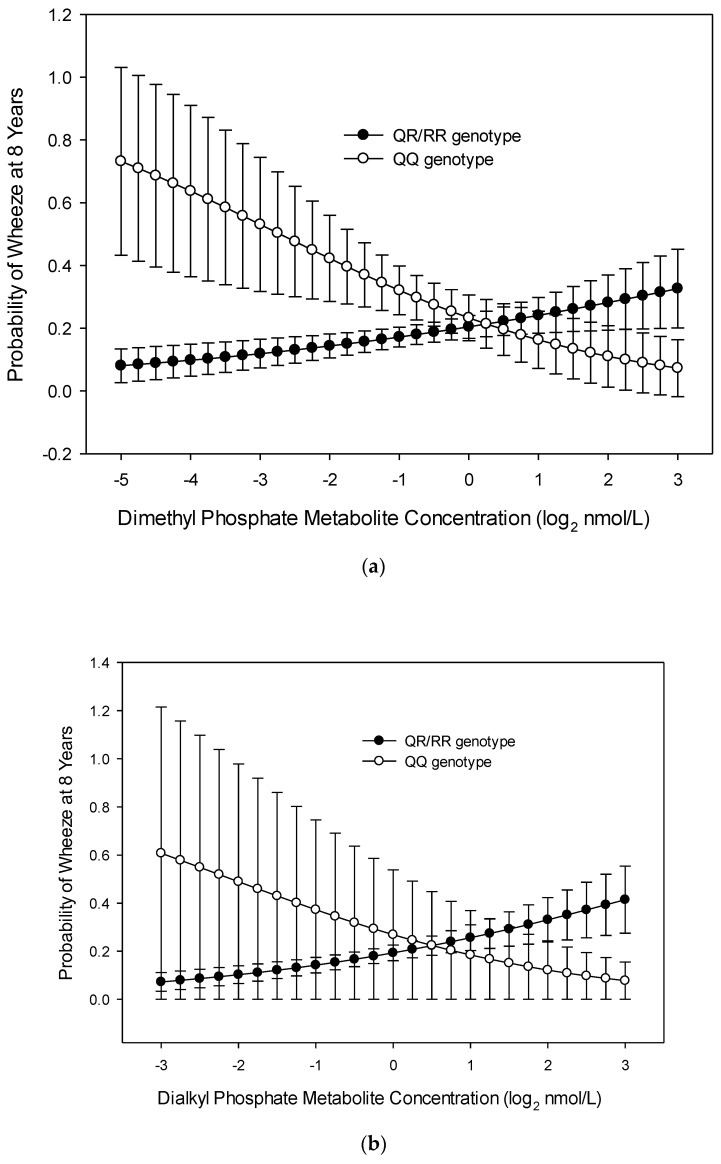
Probability of child wheeze at age 8 years related to maternal *PON1_192_* genotype and maternal (**a**) dimethyl phosphate (DM) metabolite concentration, (**b**) dialkyl phosphate (DAP) metabolite concentration, (**c**) diethyl phosphate (DE) metabolite concentration, and (**d**) pyrethroid (3-Phenoxybenzoic acid, 3PBA) metabolite concentration during gestation (*n* = 224).

**Figure 4 ijerph-17-07165-f004:**
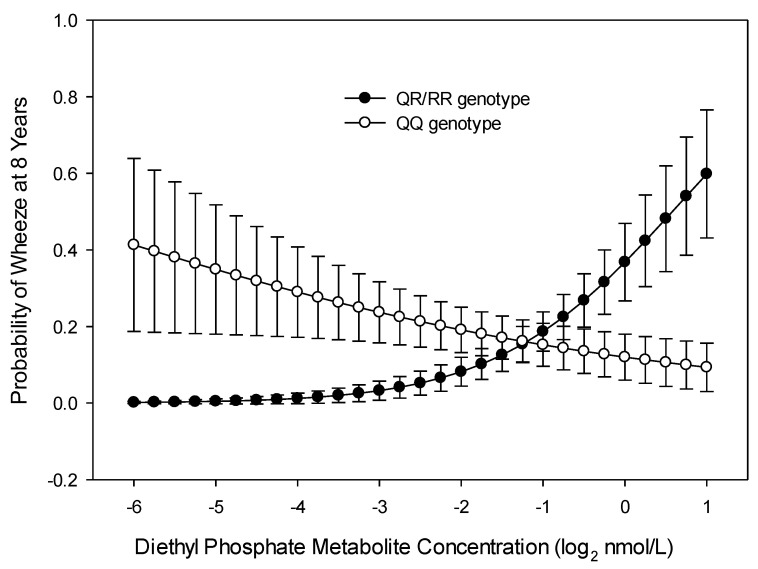
Probability of child wheeze at age 8 years related to maternal *PON1_192_* genotype and maternal diethyl phosphate (DE) metabolite concentration during gestation among white mothers (*n* = 110).

**Figure 5 ijerph-17-07165-f005:**
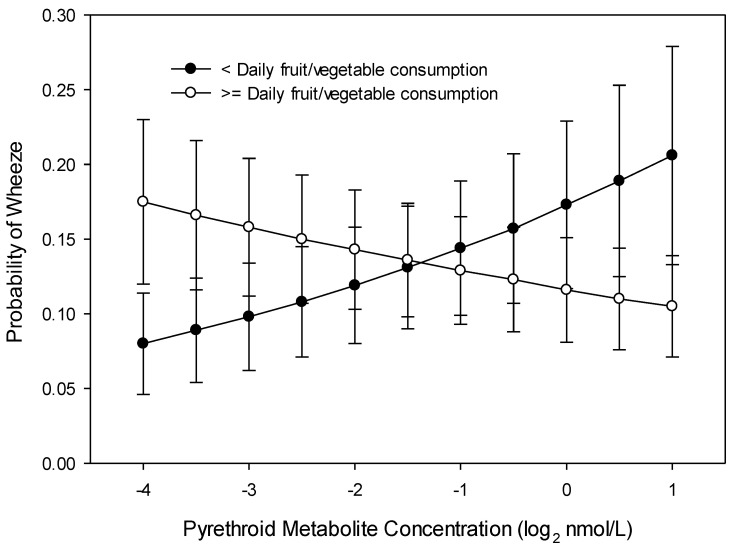
Probability of child wheeze at any age related to maternal pyrethroid (3-Phenoxybenzoic acid, 3PBA) metabolite concentration and maternal daily fruit and vegetable consumption during pregnancy (*n* = 367).

**Figure 6 ijerph-17-07165-f006:**
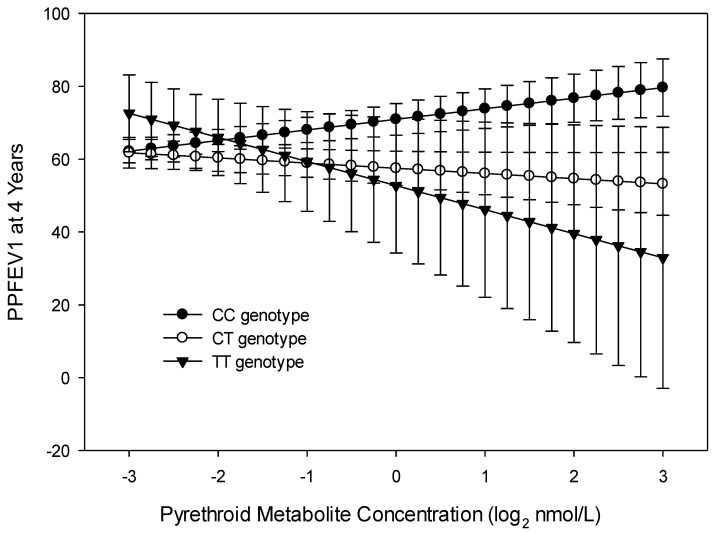
Child’s predicted percent FEV1 at age 4 years related to maternal *PON1_−108_* genotype and maternal pyrethroid (3-Phenoxybenzoic acid, 3PBA) metabolite concentration (*n* = 102).

**Table 1 ijerph-17-07165-t001:** Characteristics of participants with child wheeze data reported (N = 367).

	Wheeze (*n* = 205)		No Wheeze (*n* = 162)		All (*n* = 367)	
	*n*	%	*n*	%	*n* ^a^	%
Mother						
Race						
White, non-Hispanic	135	57.4	100	42.6	235	64.2
Others	69	52.7	62	47.3	131	35.8
Education at Delivery						
Bachelor’s degree	4	100.0	0	0.0	4	2.1
Some college or 2 year degree	59	54.1	50	45.9	109	56.8
Some high School (Grades 9–12)	39	49.4	40	50.6	79	41.1
Insurance at Delivery						
Private	150	56.4	116	43.6	266	72.7
Public/uninsured	54	54.0	46	46.0	100	27.3
Marital Status at Delivery						
Married	133	55.6	106	44.4	239	65.3
Not Married	69	54.3	58	45.7	127	34.7
Employment at Delivery						
No	37	54.4	31	45.6	68	18.6
Yes	167	56.0	131	44.0	298	81.4
Household Income at Delivery						
Less than USD 20,000	44	55.7	35	44.3	79	21.5
USD 20,000–USD 40,000	37	61.7	23	38.3	60	16.3
Greater than USD 40,000	124	54.4	104	45.6	228	62.1
Fruit/Veg Consumption During Pregnancy						
Less than daily	38	52.1	35	47.9	73	19.9
Daily	166	56.7	127	43.3	293	80.1
Parity						
0	86	53.1	76	46.9	162	44.3
1	67	57.3	50	42.7	117	32.0
>1	51	58.6	36	41.4	87	23.8
Age at Delivery (y, mean ± SD)	29.3 +/− 5.8		29.8 +/− 5.8		29.3 +/− 5.8	
Maternal Cotinine Levels at 16W (ng/mL mean ± SD)	11.6 +/− 54.7		7.7 +/− 27.1		9.5 +/− 41.8	
*PON1_−108_*						
CC ^b^	81	57.9	59	42.1	140	46.1
CT	67	53.6	58	46.4	125	41.1
TT	19	48.7	20	51.3	39	12.8
*PON1_192_*						
QQ	64	57.7	47	42.3	111	34.8
QR	75	56.8	57	43.2	132	41.4
RR	36	47.4	40	52.6	76	23.8
Child						
Race						
White, non-Hispanic	128	56.6	98	43.4	226	61.7
Others	76	54.3	64	45.7	140	38.3
Sex						
Female	109	54.8	90	45.2	199	54.4
Male	95	56.9	72	43.1	167	45.6
Birth weight (g, mean ± SD)	3390.5 ± 644.7		3381.1 ± 569.2		3359.8 ± 627.4	
Birth length (cm, mean ± SD)	50.9 ± 3.1		51.0 ± 2.7		50.8 ± 3.0	

^a^*n* is total sample size, varies due to missing data; ^b^ CC, CT, TT, QQ, QR, RR are genetic polymorphisms.

**Table 2 ijerph-17-07165-t002:** Gestational urinary pesticide metabolite concentrations by race at 16 and 26 weeks gestation.

	All (*n* = 367)			Non-white (*n* = 131)			White (*n* = 235)			Difference between Non-White and White
	Min	Max	Geometric Mean	Min	Max	Geometric Mean	Min	Max	Geometric Mean	*p*-value ^a^
3PBA 16W (ng/mL)	0.00	41.7	0.4	0.03	41.7	0.5	0.02	31.0	0.3	0.005
3PBA 26W (ng/mL)	0.00	37.4	0.3	0.03	32.2	0.4	0.02	37.4	0.3	0.002
DE 16W (nmol/L)	0.01	383.7	47.9	0.09	289.4	10.6	0.07	383.7	9.8	0.677
DE 26W (nmol/L)	0.10	594.8	28.7	0.08	594.8	6.2	0.05	231.0	5.8	0.773
DM 16W (nmol/L)	0.10	6945.9	45.2	0.38	4908.5	43.1	0.10	6945.8	46.6	0.673
DM 26W (nmol/L)	0.10	7299.3	33.0	0.72	7299.3	46.4	0.09	1606.5	26.5	0.003
DAP 16W (nmol/L)	1.30	7021.3	72.6	1.64	4921.8	72.6	1.26	7021.3	72.5	0.991
DAP 26W (nmol/L)	0.30	7894.1	53.2	1.38	7894.0	69.5	0.34	1739.1	44.8	0.008

3PBA = pyrethroid (3-Phenoxybenzoic acid), DE = diethyl phosphate, DM = dimethyl phosphate, DAP = dialkyl phosphate. ^a^ t-test *p*-value.

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
