# Peer review of "Gestational Pesticide Exposure and Child Respiratory Health"

_ijerph, 2020, doi:10.3390/ijerph17197165_

Round 1
Reviewer 1 Report
This is a decent study exploring the association between gestational pesticide exposure and child respiratory health. Although the result is good, it is a bit lengthy. The authors may want to make it concise and focus on things that are significant. For example, race-stratified analysis could be dismissed, because it does not give any different outcome.
Some questions are listed as follows:
When analyzing an independent pesticide metabolite (e.g., DM), did the authors control for other metabolites in the regression analysis?
Compared to other pesticide studies, were the concentrations of pesticide metabolites considered high, fair, low or about the same? Concentrations may determine the significance of results.
Is it necessary to conduct separate analyses for DM and DE? Since the author did not tend to explain the differences between DM and DE, DAP analyses would be just OK.
Author Response
See attached document
Reviewer 2 Report
The paper by Gilden et al. “Gestational Pesticide Exposure and Child Respiratory Health” studies the relationship of childhood asthma with maternal pesticide exposure, fruit consumption, and genetics. I have several concerns.
L24
Please explain FEV1 first before using the acronym.
L29-30
Please explain CT/TT, QR/RR, DE, DAP first.
L60
Please include more studies on the potential respiratory health benefits of eating fruits. The sole study of Mexico City does not quite justify the consideration of fruit consumption in the current paper.
L67
Please explain “secondary”.
L73
Please explain the selection of houses built before 1978. Potential health hazards contributing to asthma in such houses?
L95
Please explain the standardization procedure. How 100 is used here?
L129
A p-value of 0.05 is more used traditionally. Please explain the use of 0.2.
L156-161
I am confused about why multiple modeling approaches are used here. Do they really tell different stories? Or is it due to the data type of predictors?
L191 Table 1
I do not see serum cotinine in the table.
L195
Please include the name of the statistical test used.
Table 2
P should be explained along with the name of the statistical test.
Figure 1
Table 1 shows that the number of participants eating fruits every day is 293 while 73 for eating less than daily. Will the huge difference in subgroup sizes affect the statistical representation of the probability of wheeze?
Conclusions L217
I do not totally buy the health benefits of eating fruits to mitigate asthma. The sample size is only a few hundred. And other potential risk factors are not considered such as air pollution. The authors should try to soften the tone and revise this conclusion accordingly.
Reviewer 3 Report
In this study, the authors aimed to investigate the effect of pesticide exposure during pregnancy on respiratory health of children later on. Therefore they measured pesticide metabolites in gestational urine and correlated this data with lung function measurements and other birth cohort data from children at the age of 8 years.
This is an interesting and important study, but I do have some comments listed below that would need to be addressed. Most importantly, peadiatric wheeze is obviously not the same as asthma and actually many childhood ‘wheezers’ are transient and do not progress towards asthma. This needs to be better explained in the introduction and results. Actually, as the children are 8 years old at analysis, it should be possible to do a proper asthma diagnosis for some. Is there any data on this?
Wheezing in childhood is frequently associated with viral infections, has this been analyzed/assessed in this study? It is unclear from the methods if every child that had any wheezing episode in the 8 years is classified as wheeze? It might be an idea to stratify in ‘persistant wheezers’ vs ‘transient/one time wheezers’ as this might be a better estimate regarding respiratory health outcomes. This is very briefly mentioned in the last paragraph and that there was no association with the pesticide exposures. So in my opinion this would mean that there is no effect of pesticides on respiratory outcome, or is the power too low to conclude this ?
The assessment of the daily fruit intake needs to be explained in the methods, especially how much daily consumption constitutes a ‘yes’or ‘no’.
A strong predictor for poor child respiratory health is smoking during pregnancy. Data on smoking status needs to be shown in the table and it should be treated as a confounder in the analysis. This is a very crucial point for the results of this study.
Are there any data on exposure to air pollution (rural vs city environment)? If so please include them in the table.
Reviewer 4 Report
Dear Authors,
Thank you for the opportunity to review the manuscript titled, "Gestational Pesticide Exposure and Child Respiratory Health." The manuscript is well-developed, well-written, and addresses a thought-provoking environmental-public health issue for health providers to consider when providing health education to their patients. Overall, this is a well conducted study that should be shared with the journal's readership.
Round 2
Reviewer 2 Report
Table 1 shows that the number of participants eating fruits every day is 293 while 73 for eating less than daily. Will the huge difference in subgroup sizes affect the statistical representation of the probability of wheeze?
This questions is not answered.
Reviewer 3 Report
Thank you very much for your answers.
I would however recommend to mention the assessment of cotinine/smoking status in the manus ript amd why this was excluded as a confounder.
